# Novel Polyomaviruses in Mammals from Multiple Orders and Reassessment of Polyomavirus Evolution and Taxonomy

**DOI:** 10.3390/v11100930

**Published:** 2019-10-10

**Authors:** Bernhard Ehlers, Augustin E. Anoh, Nicole Ben Salem, Sebastian Broll, Emmanuel Couacy-Hymann, Daniela Fischer, Alma Gedvilaite, Nanina Ingenhütt, Sonja Liebmann, Maite Martin, Arsene Mossoun, Lawrence Mugisha, Jean-Jacques Muyembe-Tamfum, Maude Pauly, Bernat Pérez de Val, Hannah Preugschas, Dania Richter, Grit Schubert, Claudia A. Szentiks, Tamara Teichmann, Cornelia Walter, Rainer G. Ulrich, Lidewij Wiersma, Fabian H. Leendertz, Sébastien Calvignac-Spencer

**Affiliations:** 1Division 12 “Measles, Mumps, Rubella, and Viruses Affecting Immunocompromised Patients”, Robert Koch Institute, 13353 Berlin, Germany; 2Laboratoire de Zoologie-Biologie, UFR Biosciences Université Félix Houphouët Boigny, Abidjan BP V 34, Cote D’Ivoire; 3Centre de Recherche Pour le Développement, Université Alassane Ouattara, Bouaké BP V 1801, Cote D’Ivoire; 4Laboratoire National D’appui au Développement Agricole/Laboratoire Central de Pathologie Animale, Bingerville BP 206, Cote D’Ivoire; 5Institute of Biotechnology, Vilnius University, LT-10257 Vilnius, Lithuania; 6Institut de Recerca i Tecnologia Agroalimentàries (IRTA), Centre de Recerca en Sanitat Animal (CReSA, IRTA-UAB), 08197 Barcelona, Spain; 7EcoHealth Research Group, Conservation & Ecosystem Health Alliance (CEHA), Kampala 34153, Uganda; 8College of Veterinary Medicine, Animal Resources & Biosecurity (COVAB), Makerere University, Kampala 34153, Uganda; 9Institut National de Recherche Bio-Médicale, Kinshasa-Gombe BP 1197, Congo; 10P3 “Epidemiology of highly pathogenic microorganisms”, Robert Koch Institute, 13353 Berlin, Germany; 11Landscape Ecology & Environmental Systems Analysis, Institute of Geoecology, Technische Universität Braunschweig, 38106 Braunschweig, Germany; 12Leibniz Institute for Zoo and Wildlife Research (IZW), 10315 Berlin, Germany; 13Friedrich-Loeffler-Institut, Institute of Novel and Emerging Infectious Diseases, 17493 Greifswald-Insel Riems, Germany; 14German Center for Infection Research (DZIF), Partner Site Hamburg-Lübeck-Borstel-Insel Riems, 17493 Greifswald-Insel Riems, Germany; 15P3 “Viral Evolution”, Robert Koch Institute, 13353 Berlin, Germany

**Keywords:** polyomavirus, genome, evolution, T antigen, VP2, splicing, taxonomy

## Abstract

As the phylogenetic organization of mammalian polyomaviruses is complex and currently incompletely resolved, we aimed at a deeper insight into their evolution by identifying polyomaviruses in host orders and families that have either rarely or not been studied. Sixteen unknown and two known polyomaviruses were identified in animals that belong to 5 orders, 16 genera, and 16 species. From 11 novel polyomaviruses, full genomes could be determined. Splice sites were predicted for large and small T antigen (LTAg, STAg) coding sequences (CDS) and examined experimentally in transfected cell culture. In addition, splice sites of seven published polyomaviruses were analyzed. Based on these data, LTAg and STAg annotations were corrected for 10/86 and 74/86 published polyomaviruses, respectively. For 25 polyomaviruses, a spliced middle T CDS was observed or predicted. Splice sites that likely indicate expression of additional, alternative T antigens, were experimentally detected for six polyomaviruses. In contrast to all other mammalian polyomaviruses, three closely related cetartiodactyl polyomaviruses display two introns within their LTAg CDS. In addition, the VP2 of Glis glis (edible dormouse) polyomavirus 1 was observed to be encoded by a spliced transcript, a unique experimental finding within the *Polyomaviridae* family. Co-phylogenetic analyses based on LTAg CDS revealed a measurable signal of codivergence when considering all mammalian polyomaviruses, most likely driven by relatively recent codivergence events. Lineage duplication was the only other process whose influence on polyomavirus evolution was unambiguous. Finally, our analyses suggest that an update of the taxonomy of the family is required, including the creation of novel genera of mammalian and non-mammalian polyomaviruses.

## 1. Introduction

Polyomaviruses are DNA viruses that infect mammals, birds, and fish [1]. Polyomavirus-like sequences have also been recovered from arthropods which hints towards a very ancient association of polyomaviruses with animals [2]. The pace at which polyomaviruses have been discovered, rose enormously within the last 10 years, with >100 distinct polyomaviruses currently known, which resulted in a much better understanding of their evolution and its underlying processes (ICTV Online (10th) Report: www.ictv.global/report/polyomaviridae). The polyomaviruses infecting mammals generally appear as very host-specific, which in many cases seems to have led to virus–host codivergence ([3,4,5]; reviewed in: [1]). However, lineage duplications are also obvious because phylogenetic analyses have shown that both primate and bat polyomaviruses belong to multiple, distant monophyletic groups (e.g., [1]). Similarly, incongruences between trees reconstructed from polyomavirus’s early and late regions have been taken as evidence that recombination sometimes reshuffled the early and late regions of highly diverged genomes (from humans and bats) [6,7,8], although others have suggested that substitution rate variation may be a more likely explanation [9]. Finally, circumstantial genetic evidence for cross-species transmission of polyomaviruses between members of different bat species [10], between bats and humans [8], and shrews and humans [11] has been reported. However, this issue needs further investigation. 

Our understanding of mammalian polyomavirus evolution originates from studies that focused on samples obtained from animals belonging to only a handful of mammalian orders (Primates, Chiroptera, Rodentia, Carnivora, Cetartiodactyla, and Perissodactyla), with a heavy bias towards primates (humans, great apes, and monkeys) and bats (ICTV Online (10th) Report: www.ictv.global/report/polyomaviridae). Even within these orders, only a very limited number of species have been assessed for the presence and diversity of polyomaviruses (in total, probably fewer than 100). There are about 5000 mammalian species belonging to 29 orders. Therefore, it is evident that our current view on polyomavirus evolution stems from nothing more than a (brief) glance through a keyhole. Here, we aimed at broadening our knowledge of mammalian polyomavirus diversity, and thereby of the evolutionary processes that generated it, by investigating 1222 members of 44 mammalian (non-primate; non-bat) species belonging to 34 families and 7 orders for the presence of yet unknown polyomaviruses. For this purpose, we analyzed tissue, blood, and fecal samples from animals originating from 12 countries on 4 continents (Africa, Asia, Europe, and North America) with a generic PCR that previously allowed us to detect a large diversity of primate polyomaviruses [12,13,14,15]. In case of successful polyomavirus sequence detection we aimed at (i) characterizing full genomes including the experimental identification in cell culture of spliced mRNAs encoding T antigens, and (ii) analyzing polyomavirus evolution by phylogenetic analysis of early coding sequences and proteins. We report on the discovery of 16 novel polyomaviruses, including 11 novel putative species. We use these new sequences to reconstruct the largest mammalian polyomavirus phylogeny to date, including 97 putative polyomavirus species infecting mammals belonging to nine orders. Building on this phylogeny as well as on our and others’ experimental data on spliced mRNAs encoding T antigens, we provide a rationale for future evolution-informed annotation of mammalian polyomavirus genomes. Finally, these new data and analyses allow us to reassess the contribution of various processes to mammalian polyomavirus evolution and to formulate suggestions of taxonomic updates.

## 2. Materials and Methods 

### 2.1. Collection of Samples and DNA Extraction

Tissue and blood samples from wild duikers, red river hog, sitatunga, and water chevrotain in the Democratic Republic of the Congo and from domestic sheep, goats, and dogs in Uganda were collected from live or deceased animals and fecal swabs were taken during the course of long-term projects focused on infectious disease in wild-living and domestic animals. General permission for sample collection was obtained from the respective national authorities. Samples had already been used in previous studies [16,17]. Left-over archival materials from previous studies or from deceased individuals (naturally or traffic accident, hunt, rabies surveillance), stored at −80 °C, originated from the following animal species: domestic sheep, cattle, goats, and pigs from Côte d’Ivoire [18]; wild black-backed jackals, lions, bat-eared foxes, and spotted hyenas from Tanzania [19]; mountain zebras from Namibia [20]; wild rodents, i.e., Norway rats, Malayan field rats, greater bandicoot rats, Savile’s bandicoot rats, muskrats, Syrian hamsters, bank voles, common voles, house mice, striped field mice, wood mice, yellow-necked mice, edible dormice, garden dormice, hazel dormice, and multimammate mice, from Germany, Thailand, and Côte d’Ivoire [21,22]; domestic pigs and wild boars from Germany [23]; domestic cattle and goats from Spain (collected for diagnostics in the Spanish tuberculosis eradication campaign); alpacas and domestic pigs from Spain [24,25]), bottlenose dolphin from Germany [26]; European pole cats from Germany [27], red foxes, raccoon dogs (collected for rabies surveillance), and wolves from Germany (collected from road accidents or illegal shootings); dogs from Belgium [28]; rock hyraxes from Tierpark Berlin, Germany (post-mortem diagnostic samples); and koalas from Zoological Garden, Duisburg, Germany [29]. All mammalian species included in this study are listed with their common and taxonomic names in Table 1.

Samples were preserved in liquid nitrogen and later transferred to −80 °C at the Robert Koch-Institute, or stored directly in RNAlater (Qiagen, Hilden, Germany). From organs and blood, DNA was isolated using the DNeasy Blood & Tissue Kit (Qiagen, Hilden, Germany). From fecal samples, DNA was extracted with the GeneMATRIX stool DNA purification kit (Roboklon, Berlin, Germany).

### 2.2. PCR Methods

Generic PCR was performed using degenerate and deoxyinosine-substituted primers as described previously. The PCR amplifies a conserved region in the VP1 gene of polyomaviruses [12]. Chest cavity fluids from yellow-necked mice were analyzed with generic VP1 PCR as described by Johne et al. [30].

Specific nested PCR (primers not listed) for detection of polyomavirus sequences in 300 ng (measured with Nanodrop spectrophotometer) of domestic goat, pig, wolf, lion, common tree shrew, edible dormouse, and multimammate mouse DNA was performed in a total volume of 25 µL with 20 pmol of each primer (Metabion, Martinsried, Germany), 0.2 µL AmpliTaq Gold, 200 µM dNTPs, 2 mM MgCl_2_ (Applied Biosystems GmbH, Darmstadt, Germany), and 5% DMSO (Sigma-Aldrich Chemie GmbH, Taufkirchen, Germany). A T-Gradient thermocycler (Biometra, Jena, Germany) was used with the following cycling conditions: 95 °C for 12 min, and 45 cycles of 95 °C for 30 s, 58–60 °C (1st and 2nd round) for 30 s, and 72 °C for 2 min, followed by a 15 min final extension step at 72 °C. 

To completely amplify and sequence polyomavirus genomes (approximately 5 kilobases (kb)), nested long-distance PCR (LD-PCR) was performed with the TaKaRa Ex Taq PCR Kit (Takara Bio Europe, Saint-Germain-en-Laye, France) according to the instructions of the manufacturer. As a template, 300 ng of tissue DNA was used, and an annealing temperature of 55 °C was applied. LD-PCR was carried out with inversely oriented nested primers (Appendix A) that target the VP1 sequences generated with generic VP1 PCR. The LD fragments were sequenced with a classical primer-walking strategy and assembled with the partial VP1 sequences to full circular genomes. As for each genome, the overlaps between the VP1 sequence and the LD sequence where short, specific nested PCR (primers not listed) was performed that widely encompasses this region. The resulting sequences were assembled to final circular genome sequences.

To analyze splice events in the early region of the identified polyomaviruses, specific nested PCR (primers not listed) or LD-PCR (primers not listed) was performed under the conditions described above, using as template 2.5 µL cDNA (synthesized as described below).

### 2.3. Synthesis of Polyomavirus Early Regions and Transfection in Cell Culture

Early regions plus flanking sequences of fully amplified and sequenced polyomavirus genomes were commercially synthesized and delivered inserted in recombinant plasmids (Biomatik, Ontario, Canada). Cell lines used for the transfection of polyomavirus early regions are listed with culture conditions and transfection parameters in Appendix A. For transfection, the reagents X-treme GENE (Roche Applied Biosciences, Mannheim, Germany) and GeneJuice® (Novagen®/Merck, Darmstadt, Germany) were used. Cells were seeded in a volume of 500 µL cell culture medium in tissue culture plates with 24 wells (Nunc, Roskilde, Denmark). Transfection procedures were performed approximately 20 h after seeding according to the manufacturer’s instructions.

### 2.4. RNA Extraction and cDNA Synthesis 

RNA was isolated at 1, 2.5, and 6 days post-transfection of early region plasmid DNA using the Macherey & Nagel Total RNA Isolation Kit (Macherey & Nagel, Düren, Germany) according to the manufacturer´s instruction. DNA was removed by an additional Turbo DNA-free™ DNase treatment (Ambion, Austin TX, USA). RNA concentrations were determined with the NanoDrop™ 8000 (ThermoScientific, Waltham MA, USA) at 260 nm. Synthesis of cDNA was carried out with 500 ng RNA using SuperScript® II Reverse Transcriptase and Oligo(dT)16 primers (Invitrogen, Carlsbad CA, USA).

### 2.5. Purification of PCR Products and Sequencing

PCR products were purified using the Invisorb DNA clean up kit (Invitek, Berlin, Germany) according to the manufacturer’s instructions. Sequencing reactions were performed with the Big Dye terminator cycle sequencing kit (Applied Biosystems, Warrington, UK) and products analyzed on a 377 automated DNA sequencer (Applied Biosystems).

### 2.6. Prediction of Splice Sites in Early Regions of the Novel Polyomavirus Genomes

Before experimental analysis, potential splice donor (SD) and acceptor (SA) sites were predicted in the early region of the discovered polyomavirus genomes using the Human Splice Finder tool (http://www.umd.be/HSF3/index.html; [31]). Sites with a high rating (>70) were then compared with those of closely related, annotated polyomavirus genomes available in GenBank, preferentially those whose splice sites had been confirmed experimentally. By this, splice sites conserved in sequence and position were selected.

### 2.7. Check of Splice Sites and Coding Sequences (CDS) in Published, Annotated Polyomavirus Genomes

Based on the splice sites (i) experimentally identified here for 11 novel and 7 published polyomaviruses and (ii) experimentally identified for 10 published polyomaviruses by others [7,32,33,34,35,36,37,38,39,40,41,42,43], predicted large, middle, and small T antigen (LTAg, MTAg, and STAg) splice sites of 79 published, annotated polyomaviruses were checked in early region nucleic acid alignments of phylogenetically closely related viruses and if necessary corrected or newly selected. Criteria for selection of splice sites were conservation in position and sequence with those of the most closely related phylogenetic clade member which was experimentally examined. Based on these sites, annotation of CDS was—if necessary—corrected. In addition, genomes were checked for CDS potentially encoding ALTO or VP3 and respective CDS annotated, if they had not been annotated in the respective GenBank accession.

### 2.8. Sequence Datasets

For the 106 virus genomes included in this study (11 newly generated and 95 corresponding to published studies), we extracted the VP1 and LTAg CDS (corrected as above mentioned) using Geneious v11.1.5 [44]. For each CDS, sequences were translated into amino acids and aligned using MUSCLE [45] as implemented in Seaview v4 [46]. Conserved amino acid blocks were then selected using Gblocks (also implemented in Seaview) [47]. These alignments were further checked, and additional ambiguous positions removed manually. The final VP1 and LTAg amino acid alignments comprised of 262 and 444 positions, respectively.

### 2.9. Phylogenetic Analyses

We first ran maximum-likelihood analyses on both datasets using PhyML v3 with smart model selection (PhyML-SMS) using the Bayesian information criterion and a tree search using subtree pruning and regrafting [48,49,50]. Branch robustness was estimated using Shimodaira-Hasegawa-like approximate likelihood ratio tests (SH-like aLRT) [51]. We then ran Bayesian Markov chain Monte Carlo (BMCMC) analyses using BEAST v1.10.4 [52]. For each alignment, we used the amino acid substitution model identified by PhyML-SMS, an uncorrelated relaxed clock (lognormal) model and a speciation model (birth–death) as a tree prior. The output of multiple BMCMC runs was examined for convergence and appropriate sampling of the posterior using Tracer v1.7.1 [53], before being merged using LogCombiner v1.10.4 (distributed with BEAST). The maximum clade credibility (MCC) tree was identified from the posterior set of trees (PST) and annotated with TreeAnnotator v1.10.4 (also distributed with BEAST). Branch robustness was estimated based on their posterior probability in the PST.

### 2.10. Cophylogenetic Analyses

We used the LTAg MCC tree (entire tree or a subtree containing all alphapolyomaviruses), a host tree generated using TimeTree [54] and their tip associations to generate a tanglegram with TreeMap v3b [55]. Given the size of the virus and host trees, applying the codivergence test implemented in TreeMap v3b led to prohibitive execution times (as reported by others on similar sized datasets [56]). The degree of topological congruence and the number of events necessary to explain (reconcile) incongruences were therefore assessed using Jane version 4 [57]. Jane implements a genetic algorithm to quickly identify the most parsimonious scenarios of coevolution involving several types of events (codivergence, duplication, duplication with host switch, loss, and failure to diverge). For our analyses, we used an event cost matrix for which codivergence events were set at −1 and all non-codivergence events at 0, assuming that (i) duplication incurs costs related to within-host speciation, e.g., maintaining of distinct lineages in the face of within-host competition or tropism change within the same host, (ii) host switch incurs costs, (iii) loss is less likely than codivergence as prevalence is (and probably always was) high for most polyomaviruses, and (iv) given their respective evolutionary timescales, viruses are unlikely to fail to diverge when their hosts do so. Jane was run using the vertex-based cost mode, and the parameters of the genetic algorithm were kept at their default values (population size, 100; number of generations, 100). To determine the probability of observing the inferred costs by chance, costs were also calculated on a set of 100 samples for which tip mapping was randomized. Settings of the genetic algorithm were kept at default values. Finally, to estimate whether deep codivergence events may be suspected, we inspected all well-supported clades in the LTAg MCC tree that comprised representatives of more than two mammalian or avian orders. We used a recent genome-wide phylogeny as a reference for placental mammal phylogeny [58].

## 3. Results

### 3.1. Identification and Characterization of Polyomaviruses in Non-Hominine Mammals

To extend the knowledge of the diversity of polyomaviruses in wild, captive, and domestic mammals, 1614 blood, tissue, and fecal samples were collected from 1222 animals (44 different species from 7 orders; Table 1). They were analyzed with a generic PCR that broadly detects VP1 sequence of polyomaviruses [12,13,14]. Sixty-four (4%) samples were PCR positive and identified to originate from 18 distinct polyomaviruses as determined with sequencing and BLAST analysis of sequences. They were detected in animals belonging to 16 host species (one polyomavirus/host species): domestic cattle (*Bos taurus*), blue duiker (*Philantomba monticola*), Peters´ duiker (*Cephalophus callipygus*), bottlenose dolphin (*Tursiops truncatus*), mountain zebra (*Equus zebra zebra*), domestic pig (*Sus scrofa domesticus*), red-river hog (*Potamochoerus porcus*), wolf (*Canis lupus*), lion (*Panthera leo*), common tree shrew (*Tupaia glis*), Norway rat (*Rattus norvegicus*), house mouse (*Mus musculus*), yellow-necked mouse (*Apodemus flavicollis*), edible dormouse (*Glis glis*), and multimammate mouse (*Mastomys natalensis*). In domestic goats (*Capra aegragus hircus*), three different polyomaviruses were identified (Table 2). Of these, 2 of the 18 polyomaviruses were already known: bovine polyomavirus, infecting cattle [59], and murine pneumotropic virus, infecting house mice [60]. Their complete genomes were determined from the virus-positive samples of this study and reported earlier [61,62]. Following from this, 15 of the remaining 16 VP1 sequences exhibited less than 90% nucleic acid identity to each other, or to the corresponding region of known polyomaviruses. Only the two duiker VP1 sequences (Table 2) revealed 98% nucleic acid sequence identity. The 16 novel polyomaviruses were detected in 14 respective host species of four mammalian orders. They were provisionally named according to their host species and listed with name abbreviations in Table 2. 

For 11 of the 16 novel putative polyomaviruses, we were able to generate complete genomes (4699 bp–5338 bp), the respective hosts being blue duiker, domestic goat, domestic pig, red-river hog, wolf, lion, common tree shrew, Norway rat, yellow-necked mouse, multimammate mouse, and edible dormouse (Table 3). For 6 of the 11 polyomaviruses, >1 genome were determined (99–100% respective genome identity). Altogether, 24 complete genomes were generated. 

Open reading frame (ORF) analysis of the genomes revealed the typical polyomavirus organization, i.e., an early region, encoding the small and large T antigen (STAg, LTAg), and a counter clock-wise oriented late region, encoding the viral structural proteins VP1 and VP2, separated by the non-coding control region (NCCR). Five polyomaviruses were predicted to encode a middle T antigen (MTAg; [63]), and 10 a VP3 protein. Merkel cell polyomavirus was reported to possess an overprinting gene in its early region that encodes the so-called ALTO protein. Its start ATG is part of the TATGG motif which is conserved in most polyomaviruses and corresponds to the conserved YGS/T amino acid (aa) motif in the LTAg frame [64]. Open reading frames (>200 nt) that start at the above motif and encode a putative protein (>66 aa) with similarity to the ALTO protein, were identified in seven polyomaviruses (Appendix A). 

The predicted LTAg CDS of the 11 novel polyomaviruses revealed 53–82% pairwise nucleic acid sequence identity to those of the most closely related members of polyomavirus species (Appendix A). Among each other, the novel polyomaviruses revealed 63–77% LTAg identity. 

To further evaluate the potential host association of those of the 11 novel, completely sequenced polyomaviruses where only one or two of the tested samples were positive in generic PCR, virus genome-derived specific primers were selected and used in nested PCR for more sensitive screening. This was successful for seven of the eight tested polyomaviruses, thereby increasing the number of positive samples and individuals. Only for the porcine polyomavirus (SscrPyV1), both the generic and specific PCR assays revealed the same single sample as positive (Table 2).

### 3.2. Cell Culture-Based Identification of Splice Sites in Early Regions of the Novel Polyomavirus Genomes

From the experimentally identified partial mRNA sequences and splice donor (SD) and acceptor (SA) sites, coding sequences for LTAg, MTAg, STAg, and alternative T antigens (TAgs) were deduced, as described for each polyomavirus in detail in Appendix A. The identified TAg CDS and their splice site positions are depicted in Figure 1, Figure 2 and Figure 3 and listed in Appendix A with the cell lines in which they were detected. The predicted LTAg introns were experimentally identified for each of the 11 novel polyomaviruses, indicating LTAg expressed from two exons in 9 of 11 viruses. The LTAg of the cetartiodactyl-infecting CaegPyV1 (goat) and PporPyV1 (red-river hog) was observed to be encoded by three exons, resembling the LTAg CDS of BoPyV (BtauPyV1) [33] (Appendix A). For 6 of 11 polyomaviruses, both unspliced and spliced mRNA was detected that putatively encodes STAg. In these cases it was inferred that STAg is encoded by the spliced mRNA, as the unspliced mRNA may have also been pre-mRNA. In two cases (PporPyV1 and CaegPyV1) only unspliced mRNA was detected; spliced mRNA was neither predicted nor experimentally identified. For PmonPyV1, only spliced mRNA was identified. For MnatPyV2 and ClupPyV1, only unspliced mRNA was identified although a spliced mRNA was predicted for both (Figure 1, Figure 2 and Figure 3; Appendix A). Comparison with four closely related rodent polyomaviruses (Figure 4), however, led us to infer that MnatPyV2 STAg is encoded from spliced mRNA (see below; Appendix A), whereas for ClupPyV1 it was deduced from comparison with two closely related carnivore-infecting polyomaviruses (Figure 4) that STAg is encoded from unspliced transcript (see below; Appendix A). An MTAg intron, although predicted for 7 of 11 polyomaviruses, was only identified for AflaPyV1. For 5 of the 11 polyomaviruses, coding sequences for additional, alternative TAgs of 83 aa to 776 aa were identified (Figure 1, Figure 2 and Figure 3; Appendix A).

### 3.3. Cell Culture-Based Identification of Splice Sites in Early Regions of Selected Published Polyomavirus Genomes

To allow for a phylogeny-guided identification of similar, conserved splicing sites in polyomaviruses that had not been experimentally analyzed, we also performed experimental splicing analysis for seven polyomaviruses published previously. We selected them to maximize the diversity of mammalian polyomaviruses for which such information would be available: Equus caballus polyomavirus 1 (EcalPyV1; [65]), Human polyomavirus 6 (HPyV6; [66]), Meles meles polyomavirus 1 (MmelPyV1; [9]), Microtus arvalis polyomavirus 1 (MarvPyV1; [67]), Pan troglodytes polyomavirus 1 (PtroPyV1; [68]), Pan troglodytes polyomavirus 4 (PtroPyV4; [14]), Pan troglodytes polyomavirus 8 (PtroPyV8; [69]) (GenBank accession numbers and common virus names in Appendix A). While the LTAg CDS of five polyomaviruses were experimentally identified as originally annotated, it was longer than originally annotated in MarvPyV1 because the CDS only contains one intron, and in PtroPyV1 because the SA is located further upstream. All STAg-encoding mRNAs of the seven polyomaviruses were experimentally identified in agreement with their annotations, i.e., spliced either within CDS (EcalPyV1, HPyV6, MmelPyV1, PtroPyV1) or after the stop codon (EcalPyV1, PtroPyV4, PtroPyV8). An MTAg-encoding CDS was newly identified in PtroPyV1 (334 aa) and PtroPyV4 (327 aa). Additional novel findings: spliced CDS were identified that encode alternative TAgs of 45 aa (HPyV6; 1 intron in CDS), 79 aa (MmelPyV1; 1 intron in CDS) and 139 aa (PtroPyV8; 2 introns in CDS). An ORF >200 nt whose start corresponds to the conserved YGS/T aa motif in the LTAg frame and encodes a putative ALTO protein, was identified in HPyV6, MarvPyV1, PtroPyV1, and PtroPyV4. The results are described in detail in Text S9 and listed in Appendix A.

### 3.4. Phylogeny-Guided Identification of Splice Sites and CDS in Early Regions of Published, Annotated Polyomavirus Genomes

Splice sites and CDS in early regions of published annotated polyomavirus genomes that belong to ICTV-recognized species and for which early region splice sites and CDS have not been experimentally determined were checked in silico by comparison with those of experimentally analyzed polyomaviruses. We guided this comparison by assigning each polyomavirus genome to one of 14 arbitrarily-defined mammalian polyomavirus clades identified in our phylogenetic analyses; each of these clades contained at least one virus which had been experimentally analyzed for splicing (Figure 4). Three LTAg SD sites and six LTAg SA sites were identified in six polyomavirus genomes that differed from those annotated in the respective GenBank entries, and resulted in an LTAg sequence corrected in length and sequence (Appendix A). In addition, the predicted LTAg sequence of MglaPyV1 was longer than annotated in GenBank (Appendix A). As is the case for its experimentally analyzed sister virus MarvPyV1, the CDS displays only the first of the two introns that had been predicted earlier [67].

Of the polyomaviruses for which the STAg CDS had been annotated in silico and published as unspliced, 22 were identified as spliced, either within CDS (including stop codon) or downstream (Appendix A). Twenty-three STAg CDS were corrected in length (Appendix A). Seventeen MTAg CDS were identified that had not been annotated and published previously (Appendix A). In addition, STAg splice sites and STAg CDS were corrected for Bos taurus polyomavirus 1, and two rodent polyomaviruses.

### 3.5. Other Splice Sites and Coding Sequences in Novel Polyomavirus Genomes

The VP2 ORF of Glis glis polyomavirus 1 (GgliPyV1) appeared to be interrupted, similar to what had previously been described (but not experimentally analyzed) for Delphinus delphis polyomavirus 1 [70], the sister virus in VP1 phylogenetic analyses (see below). The genomic region of interruption was flanked by SD and SA sites that displayed a high rating in the Human Splice Finder tool. By applying an experimental approach comparable to that used here for the early region, we identified splice sites for VP2 CDS of GgliPyV1 that demarcate an intron of 74 nt (Figure 2, Appendix A). Of note in Canis familiaris polyomavirus 1, the single sister virus of Glis glis polyomavirus 1 in the Bayesian tree (Figure 4), but not in other polyomaviruses, we identified splice donor and acceptor sites with a high rating (>80) in VP2 CDS, suggesting a spliced CDS. We did not identify any other polyomavirus exhibiting an interrupted VP2 CDS.

Besides LTAg, MTAg, STAg, VP1, and VP2 that are encoded by all 11 novel polyomaviruses (Appendix A), 5 of 11 encode additional TAgs from spliced transcripts, 10 of 11 a VP3 protein and 6 of 11 a putative protein of unknown function from an ORF upstream of VP2 ORF (Appendix A). These proteins are examined in more detail in Text S10.

### 3.6. Phylogenetic Placement of the Novel Polyomaviruses

The LTAg trees using maximum likelihood (ML) and BMCMC approaches were very similar and confirmed most of the relationships identified in previous studies. In particular, the monophyly of members of the genera *Alphapolyomavirus*, *Betapolyomavirus,* and *Gammapolyomavirus* was strongly supported (Figure 4). Six of the newly discovered polyomaviruses nested within the diversity of alphapolyomaviruses. Mastomys natalensis polyomavirus 2 appeared in sistership with the already published Mesocricetus auratus polyomavirus 1, a clade which was itself in sistership with the monophyletic group formed by Rattus norvegicus polyomavirus 1 and Apodemus flavicollis polyomavirus 1. Together with Mus musculus polyomavirus 1, this clade of four polyomaviruses constituted a monophyletic group only comprising polyomaviruses infecting rodents (clade 7 in Figure 4). The novel Sus scrofa polyomavirus 1 appeared as the most basal offshoot of a large group of viruses whose relationships appeared as perfectly resolved and whose members otherwise infect primates, carnivore, and bats (clade 9 in Figure 4). The last two alphapolyomaviruses discovered in this study, Philantomba monticola polyomavirus 1 and Tupaia glis polyomavirus 1, formed a well-supported clade, in sistership with a group of primate-infecting polyomaviruses; this clade, in turn, formed a larger monophyletic group with a clade of bat-infecting polyomaviruses (altogether clade 10 in Figure 4).

The two novel polyomaviruses Panthera leo polyomavirus 1 and Glis glis polyomavirus 1 belonged to the clade formed by all betapolyomaviruses. Panthera leo polyomavirus 1 was part of a relatively large and poorly resolved monophyletic group comprising viruses infecting rodents, bats, and primates (clade 13 in Figure 4). Except for being part of the group formed by all betapolyomaviruses, Glis glis polyomavirus 1 placement was statistically uncertain; however, it appeared as the sister virus to Canis familiaris polyomavirus 1 which shared with it the peculiarity of an interrupted VP2 CDS (see above).

The last three novel polyomaviruses did not obviously belong to any previously recognized genus. Canis lupus polyomavirus 1 was found to belong to a clade comprising two other carnivore-infecting polyomaviruses. This clade was itself found to be the sister clade to the group formed by Human polyomavirus 10 and Human polyomavirus 11 (altogether clade 2 in Figure 4), whose monophyly with the two other recognized deltapolyomaviruses (Human polyomavirus 6 and Human polyomavirus 7) was not supported by our analysis. Finally, Potamochoerus porcus polyomavirus 1 and Capra aegagrus polyomavirus 1 formed a robust clade of cetartiodactyl-infecting viruses with Bos taurus polyomavirus 1, an unassigned polyomavirus species (clade 1 in Figure 4).

As previously reported, the VP1 phylogeny markedly differed from the LTAg phylogeny at deep nodes. However, at a shallower level, the placement of the 11 novel polyomaviruses in the VP1 phylogeny was in general compatible with their placement in the LTAg phylogeny, with two notable exceptions: (i) the viruses forming the clade 1 in the LTAg tree formed a robust group with the clade comprising carnivore-infecting polyomaviruses, and (ii) the Philantomba monticola polyomavirus 1 appeared to be the most basal member of the equivalent of LTAg clade 10 in the VP1 phylogeny (Figure 4; Figure 5).

### 3.7. Cophylogenetic Analysis

When we co-plotted mammalian host orders onto those phylogenies, polyomaviruses infecting hosts from the same order often appeared to be closely associated, sometimes forming relatively large clades, which were particularly visible in the LTAg tree (Appendix A). Host–virus LTAg tree tanglegrams suggested a measurable contribution of codivergence to polyomavirus evolution in these clades (i.e., in groups of polyomaviruses infecting hosts belonging to the same order), both when considering the overall polyomavirus tree (Appendix A) and only its best-resolved part, the very large clade gathering all alphapolyomaviruses (Appendix A). Despite the introduction of our 11 novel viruses, these trees were still very biased towards bat- and primate-infecting polyomaviruses, which represented 73 of the 97 mammalian polyomaviruses included in our analyses (75%). To formally assess whether the observed pattern of phylogenetic proximity of polyomaviruses infecting hosts from the same order only reflected this sampling bias, we used Jane to estimate the most parsimonious scenarios of host–virus association dynamics that could have generated it. For the entire tree, all most parsimonious solutions required 60 cospeciation events, while all most parsimonious solutions for the alphapolyomavirus tree required 28 such events (Table 4). In both cases, tip association randomization tests demonstrated an excess of cospeciation events in the original dataset. While a certain degree of codivergence of polyomaviruses with their hosts seemed clear within host orders (intraordinal diversification), we also wanted to determine whether deeper splits within the polyomavirus phylogeny could correspond to codivergence events at times of host order splits (putative ordinal diversification). To do this, we examined well-supported parts of the LTAg and VP1 trees comprising polyomaviruses infecting mammals from more than two orders but we failed to find any deep codivergence pattern in the LTAg tree and only found one such instance in the VP1 tree (Appendix A).

## 4. Discussion

In this study, we analyzed 1614 samples from 1222 animals of 44 mammalian species, which revealed 16 unknown and 2 known polyomaviruses in animals that belong to 5 orders (Cetartiodactyla, Perissodactyla, Rodentia, Carnivora, Scandentia), 16 genera, and 16 species. For 1 order, 7 mammalian genera, and 14 species the discovered polyomavirus was the first one. We were able to determine full genomes from 11 novel polyomaviruses, among which eight were identified with generic and specific PCR in >1 individual, thus putting strength to the animal of sequence origin representing the natural host of the respective virus. In contrast, we did not detect any polyomavirus in samples from 28 other mammalian species. These species, however, were characterized by a smaller sampling size (on average 12 samples) than the host species in which we detected polyomaviruses (on average 56 samples). Provided a large enough number of specimens is analyzed, it seems likely that most mammalian species will appear to be infected with one or several polyomaviruses, provided that the sampling includes targets the loci of tropism. This discovery effort allowed us to slightly mitigate the current host sampling bias in the mammalian polyomavirus tree: we added 11 non-primate non-bat polyomaviruses to the 23 already known, resulting in a nearly 50% increase. This prompted us to re-examine both the processes that shaped the evolution of mammalian polyomaviruses and the current taxonomy of these viruses.

Given the very high divergence of polyomavirus nucleotide sequences considered at this taxonomic scale, we and most authors in general use aa sequences in phylogenetic analyses. The longest CDS in the early region (LTAg CDS) is also frequently employed in such evolutionary analyses and the marker selected by the ICTV Polyomaviridae study group to orient sequence-based taxonomy [71]. Since this CDS is also interrupted by an intron (introns), the proper identification of splicing sites is a key determinant of the quality of all downstream evolutionary and taxonomic analyses. Therefore, we first determined TAg CDS experimentally in cell culture. Spliced mRNA sequences and the respective splice sites were identified for all novel polyomaviruses in 1–11 transfected cell lines. Although the predicted LTAg CDS and spliced and/or unspliced STAg CDS were experimentally identified for each novel polyomavirus, only two of the MTAg CDS predicted for eight of the novel polyomaviruses were identified. The reason may be low abundance of MTAg transcripts and the fact that the position and length of the STAg intron are very similar in three of six polyomaviruses with predicted MTAg intron (TgliPyV1, Figure 1; RnorPyV1 and MnatPyV1, Figure 2; Appendix A). This may lead to out-competition of the MTAg PCR product by the STAg PCR product in intron-spanning PCR, particularly, when MTAg mRNA is of less abundance then STAg mRNA. In addition, the failure to detect MTAg transcripts could be an indication that these viruses resemble the organization of MCPyV/ALTO instead of murine polyomavirus 1/MTAg. For these novel viruses (except ClupPyV1) an ALTO ORF was indeed predicted which theoretically resembles the second exon of MTAg. 

CDS for additional, alternative TAgs were only detected for five novel polyomaviruses. Especially these sequences, rather than those of LTAg or STAg CDS, were identified in only one cell line or in a subset of tested cell lines (Appendix A). Thus, the chance of detecting an alternative TAg CDS seems to be higher when a larger panel of cell lines is tested, as e.g., was the case for SscrPyV1 and RnorPyV1 (Figure 1, Figure 2 and Figure 3; Appendix A; Text S5). In addition, there was not always a cell line of the host species available from which the respective virus originated. 

For most of the identified CDS, the stop codon was part of the amplified and sequenced PCR product. However, this was not the case for all LTAg CDS and the two super large T CDS (encoding 776T of PmonPyV1 and 713T of PleoPyV1). For LTAg CDS, it is known from well-studied polyomaviruses (e.g., SV40 [72]) that the LTAg is encoded by a CDS with two exons, the second exon starting at a position around nucleotide 600 and terminating at the end of the early region. For the polyomaviruses reported here, we inferred the same LTAg CDS organization. Similarly, the intron/exon organization of MTAg CDS was deduced from those of mouse (MmusPyV1) and hamster (MaurPyV1) polyomavirus [43,73] and TSPyV (HuPyV8) [42].

Few of the alternative TAgs presently known (from SV40 (MmulPyV1), BK polyomavirus (HuPyV1), JC polyomavirus (HuPyV2), and MCPyV (HuPyV5)) have been analyzed in terms of their function. They were shown to have regulatory functions and to play an important role in transforming and immortalizing host cells [34,35,39,74]. According to our findings, the expression of alternative TAgs seems to be a common feature among the mammalian polyomaviruses. This is strengthened by the observation that most of the alternative TAg CDS identified here are conserved in closely related viruses (to be published elsewhere). An example published previously is given by 145T of Human polyomavirus 9 (HuPyV9) and LTAg´ and STAg´ of lymphotropic polyomavirus (of African green monkey; belongs to species *Human polyomavirus 9*) which appear to be conserved in these two viruses and five other closely related primate polyomaviruses populating the same phylogenetic clade [75]. Taking previous and current findings together, alternative TAgs likely play a role in the replication life cycles of mammalian polyomaviruses.

Using the newly identified splice sites in a phylogeny-guided effort to check already published genome annotations, we corrected a large number of LTAg and STAg CDS and also identified some MTAg that had not been annotated before. Although systematic experimental validation of splicing sites is probably not necessary, such investigations will certainly be warranted wherever novel divergent clades of polyomaviruses are discovered.

Based on this corrected dataset of LTAg CDS, we ran cophylogenetic analyses which demonstrated that codivergence was a significant contributor to the diversification of mammalian polyomaviruses, which confirms previous reports based on the analysis of smaller datasets [2,5,9]. The fact that there is no obvious codivergence event when considering deep nodes of the host phylogeny (i.e., ordinal relationships) suggests that most of the codivergence signal stems from relatively recent events. De facto, recent studies that focused on the description of the evolutionary processes acting at much shallower scales, e.g., at host family or genus scale, have often identified a very clear signature of frequent polyomavirus codivergence with their hosts [4,11]. The lack of a signal for codivergence at deeper nodes may have several, not mutually exclusive, technical and biological explanations. For example, our ability to reconstruct ancient polyomavirus divergence events is likely affected by the combined effects of strong, long-term purifying selection and saturation [76,77,78]. It may also be that ancient host divergence events happened so rapidly that they left no measurable trace in polyomavirus genomes. For example, the phylogenetic placement of the order Scandentia within Euarchontoglires is still very uncertain, despite the availability of genome-scale information [58]. Finally, it is clear that the larger the timescales considered, the more likely other processes will have been at play—which will all result in disturbing a strict codivergence pattern.

We found that none of these polyomaviruses was ever detected in another host species (and vice versa no already known polyomavirus detected in our new sample set had been initially reported from a different species), which reinforces the notion of a strong host-specificity and rare host switches in polyomaviruses (although well-documented exceptions exist; [10]). Similarly, recombination between the early and late regions of the novel polyomavirus can only be suspected in one of the 11 viruses (Philantomba monticola polyomavirus 1) and the evidence is slim given the relative lack of resolution in the relevant part of the late region tree. This seems to be in line with low recombination rates in polyomaviruses [6], for which even the existence of ancient recombination events is somehow controversial [9]. A much more readily accepted idea is that polyomaviruses have diversified faster than their hosts: lineage duplications, which may represent adaptation to new niches (e.g., new tropism), or simply long-term independent evolution of lineages adapted to the same niche, have happened relatively frequently during polyomavirus evolution. The general interspersion of polyomaviruses infecting densely sampled mammalian orders supports this notion as does the observation that the discovery of a novel virus in sparsely sampled host species often results in identifying an only distantly related virus, as exemplified by the discovery of Mastomys natalensis polyomavirus 2 in this study which is an alphapolyomavirus, whereas Mastomys natalensis polyomavirus 1 is a betapolyomavirus. Interestingly, both alpha- and beta-polyomaviruses have been identified from the four most sampled mammalian orders (Carnivora—9 polyomaviruses, Chiroptera—24 polyomaviruses, Primates—39 polyomaviruses, and Rodentia—12 polyomaviruses), the five remaining mammalian orders having delivered only alpha- or beta-polyomaviral sequences sharing the characteristic of a low sample size (altogether 13 polyomaviruses). It is tempting to hypothesize that the last common ancestor of all placental mammals was already infected with at least two polyomaviruses, which gave rise to the alpha- and beta-polyomaviruses. While lineage extinction necessarily occurred over the very long history of coevolution of mammals and their polyomaviruses, the frequency of this process is very hard to quantify when a heavy host sampling bias exists. Since polyomaviruses usually exhibit lifelong persistence and reach high prevalence in their host populations, one would, however, predict relatively rare independent extinction events (as opposed to coextinction events).

The increased sampling of the overall polyomavirus diversity results in a better understanding of the long-term dynamics of the coevolution with their hosts, which in turn can be leveraged to nurture their taxonomy. Over the last few years, the taxonomy of the family Polyomaviridae has considerably evolved with a total of 98 species recognized by the International Committee on Taxonomy of Viruses and the creation of four genera in 2015 [71]. Here, the application of the ICTV criteria for polyomavirus species delineation would result in creating 11 additional species. Our analyses including these new viruses further confirm the robustness of the genera *Alphapolyomavirus*, *Betapolyomavirus,* and *Gammapolyomavirus*. On the contrary, members of the genus *Deltapolyomavirus* do not form a strongly supported monophyletic group, as already previously observed [1]. The *Human polyomaviruses 6* and *7* (node 3) indeed appear as an evolutionary lineage independent of the clade comprising *Human polyomaviruses 10* and *11* and three polyomaviruses infecting carnivores (node 2). A possible solution to maintain monophyletic genera may be to create another genus (tentatively named here *Epsilonpolyomavirus*) to which *Human polyomaviruses 10* and *11* will be reassigned and which will also comprise the three above mentioned carnivore-infecting polyomaviruses (Figure 4). We also identified two other deep branching lineages that would be amenable for the creation of two additional genera: (i) a clade comprising *Bos taurus polyomavirus 1* and two novel cetartiodactyl-infecting polyomaviruses (node 1, tentative genus name *Zetapolyomavirus*), and (ii) Delphinus delphis polyomavirus 1 which would be the only species assigned to the tentative genus *Etapolyomavirus*. This scheme would allow that all mammalian polyomavirus species are assigned to a genus, following the current ICTV guidelines (New Rule 3.24. ratified by the ICTV in 2018 [79]). Such a taxonomy update should be extended to also assign currently unclassified invertebrate and fish polyomaviruses to genera of their own and possibly create higher taxa [80].

## Figures and Tables

**Figure 1 viruses-11-00930-f001:**
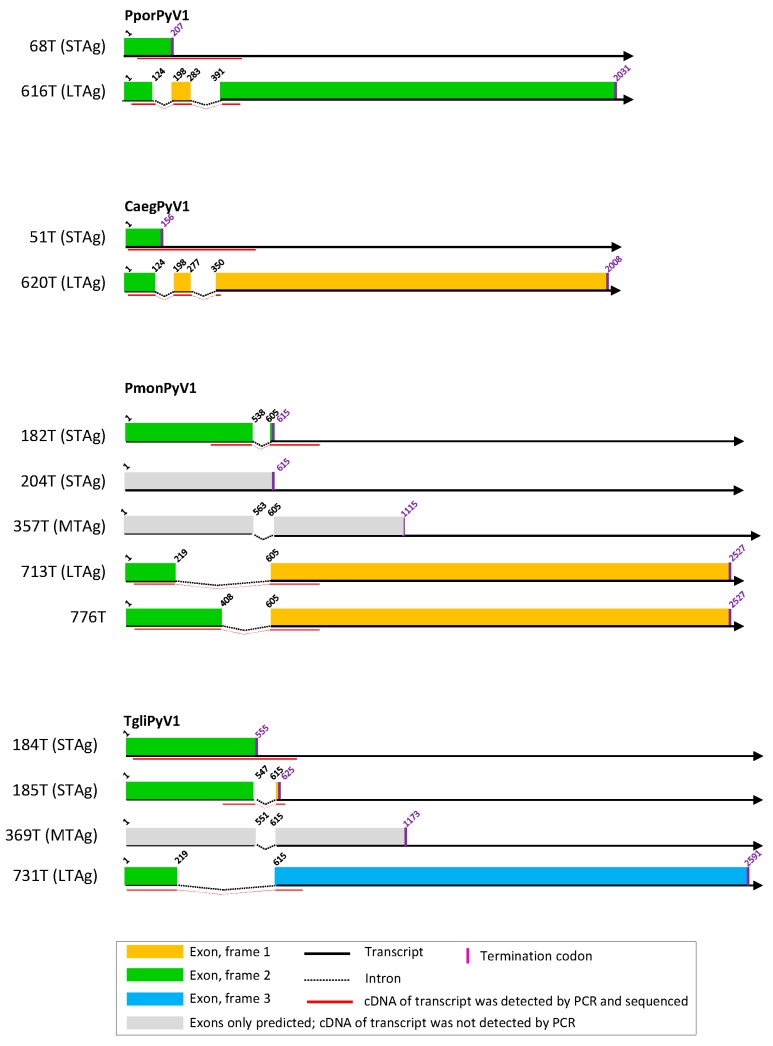
T antigen splice variants of CaegPyV1, PmonPyV1, and TgliPyV1. Experimentally identified small, middle, and large T antigens (STAg, MTag, and LTAg), and alternative TAg-encoding sequences are depicted as colored bars. STAg-, MTag-, and LTAg-encoding sequences that were predicted but not experimentally identified are depicted as gray bars. See explanatory box in the figure.

**Figure 2 viruses-11-00930-f002:**
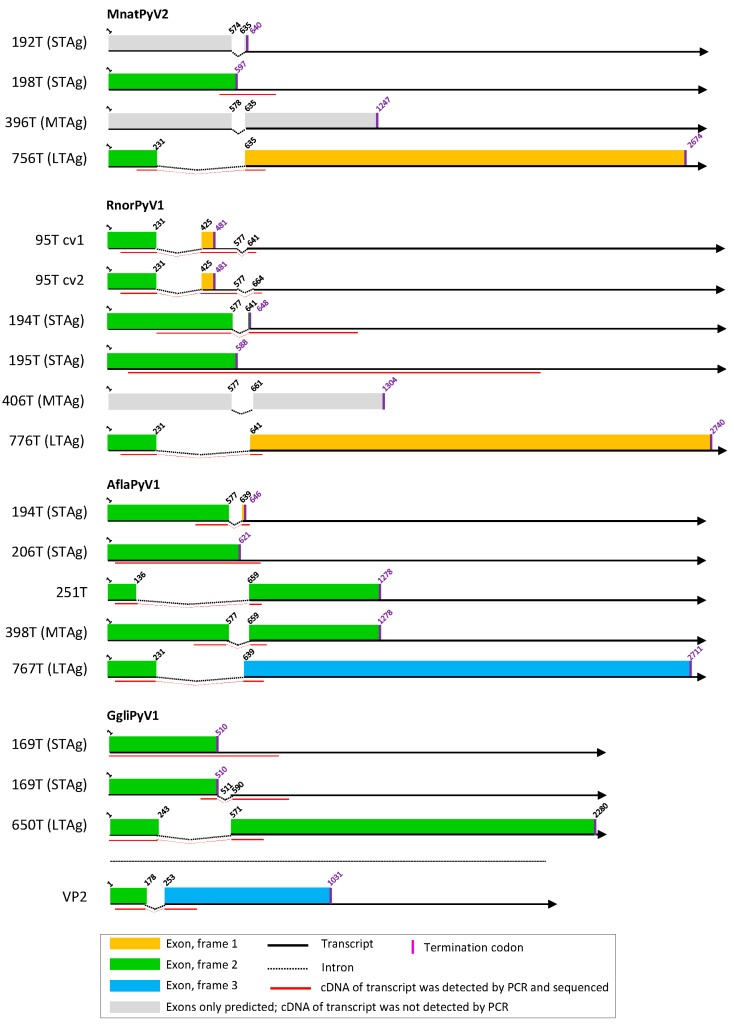
T antigen splice variants of MnatPyV2, RnorPyV1, AflaPyV1, and GgliPyV1. See legend of Figure 1.

**Figure 3 viruses-11-00930-f003:**
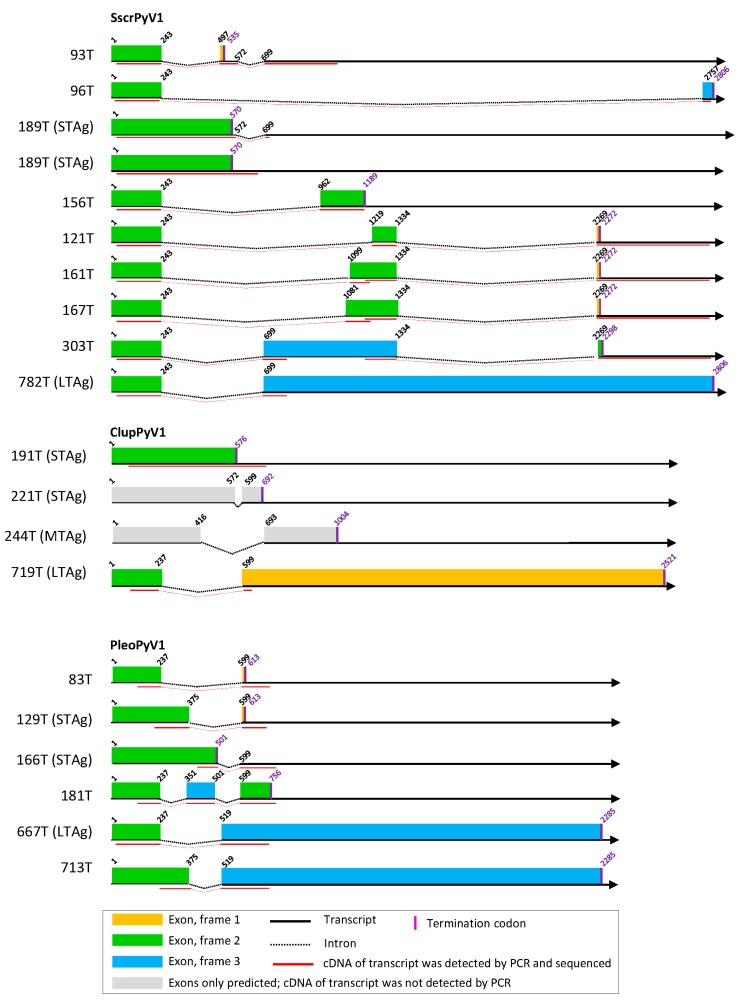
T antigen splice variants of SscrPyV1, ClupPyV1, and PleoPyV1. See legend of Figure 1.

**Figure 4 viruses-11-00930-f004:**
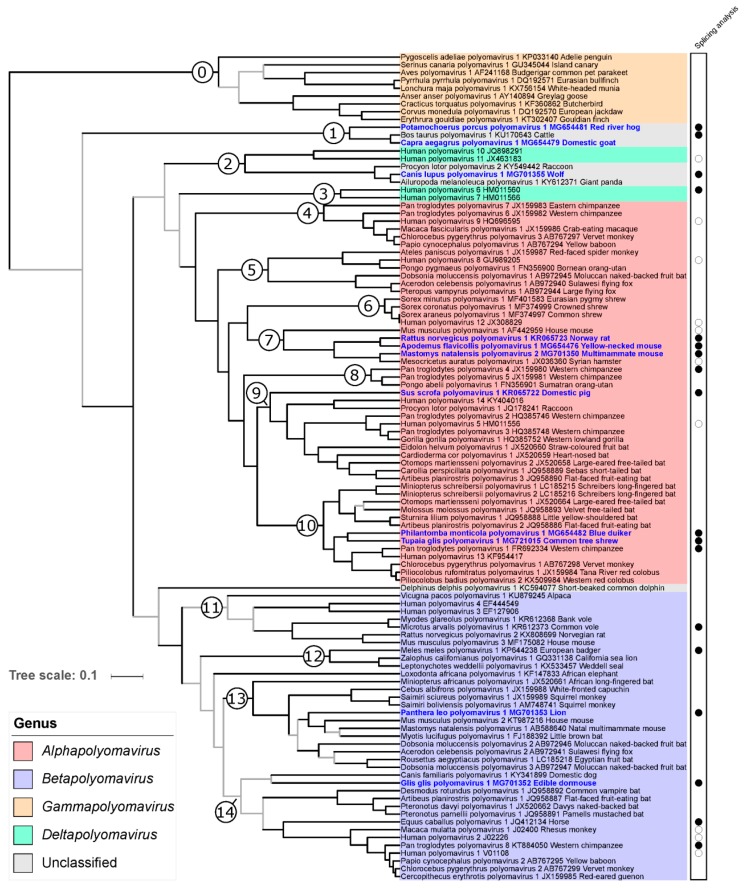
Relationships of polyomaviruses based on conserved amino acid blocks of the LTAg sequences. Polyomavirus naming follows the recommendations of the ICTV Polyomaviridae study group using Latin binomials of their hosts and a serial number; accession numbers and vernacular names of the host are also given. Virus genera are indicated by colored background. Polyomaviruses identified in this study are given in bold blue font. This maximum clade credibility tree was generated using Bayesian Markov chain Monte Carlo analyses; a maximum likelihood analysis recovered a very similar topology. Grey branches are relatively weakly supported with posterior probability values <0.95. The dataset column at the right of the tree highlights polyomaviruses for which experimental analyses of splicing sites have been performed in this or other studies (filled and empty circles, respectively). The numbered branches define clades discussed in the main text.

**Figure 5 viruses-11-00930-f005:**
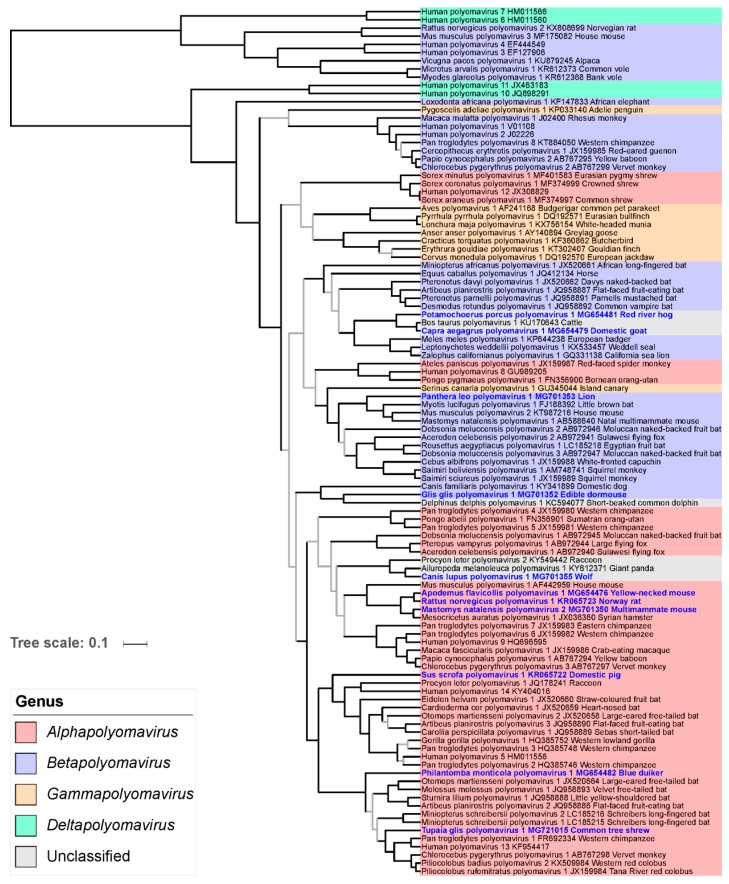
Relationships of polyomaviruses based on conserved amino acid blocks of the VP1 sequences. Polyomaviruses identified in this study are given in bold blue font. For further details see legend of Figure 4.

**Table 1 viruses-11-00930-t001:** Mammalian species tested for the presence of polyomaviruses.

Number	Host Common Name	Host Taxonomic Name	Higher Host Taxon	n Animals (Samples) Tested with Generic PCR ^a^	Organs Tested with Generic PCR	Countries of Origin of Animals Tested	Polyomavirus Positive in Generic PCR ^b^
1	Domestic cattle	*Bos taurus*	Artiodactyla	239 (247)	feces, lung, lymph node, spleen	Côte d’Ivoire, Spain, Uganda	+
2	Blue duiker	*Philantomba monticola*	Artiodactyla	19 (19)	feces, intestine, spleen	Democratic Republic of the Congo	+
3	Black-fronted duiker	*Cephalophus nigrifrons*	Artiodactyla	2 (3)	intestine, lung, spleen	Democratic Republic of the Congo	-
4	Peters´ duiker	*Cephalophus callipygus*	Artiodactyla	8 (12)	feces, intestine, lung, spleen	Democratic Republic of the Congo	+
5	Yellow-backed duiker	*Cephalophus silvicultor*	Artiodactyla	1 (1)	spleen	Democratic Republic of the Congo	-
6	Domestic goat	*Capra aegagrus hircus*	Artiodactyla	148 (159)	feces, pooled lymph nodes	Côte d’Ivoire, Spain, Uganda	+
7	Domestic pig	*Sus scrofa domesticus*	Artiodactyla	173 (359)	blood and diverse organs	Belgium, Côte d’Ivoire, Germany, Spain, Sweden, Switzerland, Uganda	+
8	Wild boar	*Sus scrofa*	Artiodactyla	22 (52)	bladder, bone marrow, spleen, tonsil	Germany	-
9	Red river hog	*Potamochoerus porcus*	Artiodactyla	1 (1)	spleen	Democratic Republic of the Congo	+
10	Sitatunga	*Tragelaphus spekii*	Artiodactyla	1 (1)	intestine	Democratic Republic of the Congo	-
11	Water chevrotein	*Hyemoschus aquaticus*	Artiodactyla	4 (8)	feces, intestine, lung, spleen	Democratic Republic of the Congo	-
12	Alpaca	*Vicugna pacos*	Artiodactyla	3 (3)	liver, lung, lymph node	Spain	-
13	Domestic sheep	*Ovis aries*	Artiodactyla	52 (52)	feces	Côte d’Ivoire, Uganda	-
14	Bottlenose dolphin	*Tursiops truncatus*	Artiodactyla	2 (6)	kidney, liver, lung, skin, spleen	Germany	+
15	Mountain zebra	*Equus zebra*	Persissodactyla	12 (12)	blood	Namibia	+
16	Domestic dog	*Canis familiaris*	Carnivora	33 (36)	blood, feces, lung, salivary gland, spleen	Germany, USA, Uganda, Belgium	-
17	Wolf	*Canis lupus*	Carnivora	49 (103)	salivary gland, spleen	Germany	+
18	Black-backed jackal	*Canis mesomelas*	Carnivora	3 (3)	lung	Tanzania	-
19	Lion	*Panthera leo*	Carnivora	28 (28)	blood, lung	Tanzania	+
20	Bat-eared fox	*Otocyon megalotis*	Carnivora	3 (3)	lung	Tanzania	-
21	Spotted hyena	*Crocuta crocuta*	Carnivora	6 (6)	blood, lung	Tanzania	-
22	Red fox	*Vulpes vulpes*	Carnivora	11 (11)	spleen	Germany	-
23	European polecat	*Mustela putorius*	Carnivora	3 (3)	spleen	Germany	-
24	Raccoon dog	*Nyctereutes procyonoides*	Carnivora	20 (20)	spleen	Germany	-
25	Common tree shrew	*Tupaia glis*	Scandentia	4 (12)	blood, lung	Thailand	+
26	Black rat	*Rattus rattus*	Rodentia	5 (5)	spleen	Thailand	-
27	Norway rat	*Rattus norvegicus*	Rodentia	33 (33)	spleen	Germany	+
28	Malayan field rat	*Rattus tiomanicus*	Rodentia	7 (7)	spleen	Germany	-
29	Greater bandicoot rat	*Bandicota indica*	Rodentia	13 (19)	lymph node, spleen	Germany	-
30	Savile’s bandicoot rat	*Bandicota savilei*	Rodentia	6 (6)	spleen	Germany	-
31	House mouse	*Mus musculus*	Rodentia	57 (76)	spleen, lung	Germany	+
32	Striped field mouse	*Apodemus agrarius*	Rodentia	12 (29)	heart, kidney, liver, lung, lymph node, spleen	Germany	-
33	Wood mouse	*Apodemus sylvaticus*	Rodentia	23 (26)	kidney, lung, spleen	Germany	-
34	Yellow-necked mouse	*Apodemus flavicollis*	Rodentia	77 (81)	chest cavity fluid, lung, spleen	Germany	+
35	Bank vole	*Myodes glareolus*	Rodentia	19 (23)	kidney, lung, lymph node, spleen	Germany	-
36	Multimammate mouse	*Mastomys natalensis*	Rodentia	49 (59)	lung, spleen	Côte d’Ivoire	+
37	Edible dormouse	*Glis glis*	Rodentia	3 (6)	spleen	Germany	+
38	Garden dormouse	*Eliomys quercinus*	Rodentia	3 (5)	kidney, spleen	Germany	-
39	Hazel dormouse	*Muscardinus avellanarius*	Rodentia	3 (4)	kidney, spleen	Germany	-
40	Common vole	*Microtus arvalis*	Rodentia	30 (36)	kidney, lung, lymph node, spleen	Germany	-
41	Muskrat	*Ondatra zibethicus*	Rodentia	19 (19)	lymph node	Germany	-
42	Syrian hamster	*Mesocricetus auratus*	Rodentia	8 (8)	spleen	Germany	-
43	Koala	*Phascolarctos cinereus*	Diprotodontia	6 (6)	blood	Germany	-
44	Rock hyrax	*Procavia capensis*	Hyracoidea	2 (6)	liver, nervus axillaris, esophagus, parotid gland, spleen	Germany	-

^a^ generic PCR as published by [1]; chest cavity fluid samples of yellow-necked mice were tested with generic PCR as published by [2]. ^b^ one or more samples positive

**Table 2 viruses-11-00930-t002:** Polyomaviruses identified.

Number	Host Common Name	Name of Identified Polyomavirus (Abbreviation)	n Samples Positive in Generic PCR (Body Compartment)	n Samples Positive in Specific PCR (Body Compartment)	n Animals Positive in Generic or Specific PCR	Country of Origin of PCR Positive Samples	Known or Novel Polyomavirus
1	Domestic cattle	Bovine polyomavirus (BoPyV)	1 (lymph node)	not done	1	Spain	known
2	Blue duiker	Philantomba monticola polyomavirus 1 (PmonPyV1)	4 (spleen)	not done	4	Democratic Republic of the Congo	novel
3	Peters´ duiker	Cephalophus callipygus polyomavirus 1 (CcalPyV1)	1 (intestine)	not done	1	Democratic Republic of the Congo	novel
4	Domestic goat	Capra aegagrus polyomavirus 1 (CaegPyV1)	1 (pooled lymph nodes)	10 (pooled lymph nodes)	6	Spain	novel
5	Domestic goat	Capra aegagrus polyomavirus 2 (CaegPyV2)	2 (pooled lymph nodes)	not done	2	Spain	novel
6	Domestic goat	Capra aegagrus polyomavirus 3 (CaegPyV3)	1 (feces)	not done	1	Uganda	novel
7	Bottlenose dolphin	Tursiops truncatus polyomavirus 1 (TtruPyV1)	1 (spleen)	not done	1	Germany	novel
8	Mountain zebra	Equus zebra polyomavirus 1 (EzebPyV1)	1 (blood)	not done	1	Namibia	novel
9	Domestic pig	Sus scrofa polyomavirus 1 (SscrPyV1)	1 (spleen)	1 (spleen)	1	Germany	novel
10	Red river hog	Potamochoerus porcus polyomavirus 1 (PporPyV1)	1 (spleen)	not done	1	Democratic Republic of the Congo	novel
11	Wolf	Canis lupus polyomavirus 1 (ClupPyV1)	1 (spleen)	5 (spleen, blood, pancreas)	4	Germany	novel
12	Lion	Panthera leo polyomavirus 1 (PleoPyV1)	1 (lung)	2 (lung)	2	Tanzania	novel
13	Common tree shrew	Tupaia glis polyomavirus 1 (TgliPyV1)	1 (spleen)	4 (spleen, lymph node)	2	Thailand	novel
14	Norway rat	Rattus norvegicus polyomavirus 1 (RnorPyV1)	22 (spleen)	not done	22	Germany	novel
15	House mouse	Murine pneumotropic virus (MPtV)	3 (spleen)	not done	3	Germany	known
16	Yellow-necked mouse	Apodemus flavicollis polyomavirus 1 (AflaPyV1)	9 (lung, chest cavity fluid)	16 (lung, chest cavity fluid)	17	Germany	novel
17	Edible dormouse	Glis glis polyomavirus 1 (GgliPyV1)	1 (spleen)	2 (spleen, kidney)	1	Germany	novel
18	Multimammate mouse	Mastomys natalensis polyomavirus 2 (MnatPyV2)	2 (lung)	1 (lung)	3	Côte d’Ivoire	novel

**Table 3 viruses-11-00930-t003:** Complete genomes of novel polyomaviruses.

Number	Name of Identified Polyomavirus	n Genomes (Sample ID)	n Animals with Full Genome Detected	Genome Length (bp)	Early Region mRNA Splice Sites Identified in Cell Culture	GenBank Accession Numbers
1	Philantomba monticola polyomavirus 1	1 (#9781)	1	5034	+	MG654482
2	Capra aegagrus polyomavirus 1	2 (#7515, #9483)	2	4699	+	MG654479, MG654480
3	Sus scrofa polyomavirus 1	1 (#0471)	1	5058	+	KR065722
4	Potamochoerus porcus polyomavirus 1	1 (#9780)	1	4825	+	MG654481
5	Canis lupus polyomavirus 1	2 (#8472; #8476)	2	5215	+	MG701355, MG701356
6	Panthera leo polyomavirus 1	2 (#3884; #3887)	2	5018	+	MG701353, MG701354
7	Tupaia glis polyomavirus 1	3 (#4373; #4376; #4472)	3	5234	+	MG721015, MG721016, MG721017
8	Rattus norvegicus polyomavirus 1	6 (#3671; #3687; #3690; #5700; #5704; #5709)	6	5318	+	KR065723, KR065724, KR075943, KR075944, KR075945, KR075946
9	Apodemus flavicollis polyomavirus 1	3 (#3349; #4021; #9779)	3	5327	+	MG654476, MG654477, MG654478
10	Mastomys natalensis polyomavirus 2	2 (8173; #8174)	2	5322	+	MG701350, MG701351
11	Glis glis polyomavirus 1	1 (#3327)	1	5338	+ ^a^	MG701352

^a^ for GgliPyV1, splice sites were identified in early and late region.

**Table 4 viruses-11-00930-t004:** Cophylogenetic analyses of host–polyomavirus evolution.

Analysis ^a^	# Cospeciations	# Duplications	# Duplications and Host Switches	# Losses	# Failures to Diverge	Cost	% Samples Best Cost ≤ Original
**Terrestrial vertebrate poylomaviruses (8679 solutions—all same costs) ^b^**	60	15	30	208	0	−60	0
**Alphapolyomaviruses (9795 solutions—all same costs) ^c^**	28	8	12	81	0	−28	0

^a^ These results were obtained using Jane version 4 and an event cost matrix were a codivergence event costs -1 and all other events cost 0. ^b^ For the terrestrial vertebrate polyomavirus analysis, 8679 solutions with the same cost were found. ^c^ For the alphapolyomavirus analysis, 9795 equally costly solutions were identified.

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
