# Peer review of "Novel Polyomaviruses in Mammals from Multiple Orders and Reassessment of Polyomavirus Evolution and Taxonomy"

_viruses, 2019, doi:10.3390/v11100930_

Round 1

Reviewer 1 Report

Ehlers and colleagues report their discovery of more than a dozen new polyomaviruses, most of which are from previously under-sampled mammalian orders. In addition to comprehensive bioinformatic analysis, the study includes wet-bench experiments addressing the splicing of the tumor antigen genes. The methods are appropriate and key claims seem supported. Overall, the results do not suggest any big theoretical surprises. The study represents a significant – albeit perhaps somewhat incremental - scientific advance.

Minor points:

Tables 6 and 7 seem too large for main text – it would be appropriate for them to be supplemental.

In Figure 4, it would be helpful to include the traditional names of human polyomaviruses – perhaps in parentheses after the accession number. Also note that it is unclear whether the virus initially dubbed “Lyon-IARC polyomavirus” should be re-named “Human polyomavirus 14” https://www.ncbi.nlm.nih.gov/pubmed/31320414 .

Lines 458-467 will be confusing to some readers. At face value it seems to first say that extensive cospeciation was detected but then tests for deep splits in host phylogeny did not detect any codivergence patterns. The authors should find a clearer way to explain the results. Discussion text starting at line 554 is somewhat clearer.

Line 518: Merkel cell polyomavirus lacks a MTAg. Instead, its ALTO gene – which theoretically resembles the second exon of MTAg – is thought to be translated through an internal ribosome entry or translational reinitiation event. The failure to detect devoted MTAg transcripts could simply be an indication that the virus resembles the organization of MCPyV/ALTO instead of murine polyomavirus 1/MTAg.

Author Response

VIRUSES_594523

Reviewer 1

Comments and Suggestions for Authors

Ehlers and colleagues report their discovery of more than a dozen new polyomaviruses, most of which are from previously under-sampled mammalian orders. In addition to comprehensive bioinformatic analysis, the study includes wet-bench experiments addressing the splicing of the tumor antigen genes. The methods are appropriate and key claims seem supported. Overall, the results do not suggest any big theoretical surprises. The study represents a significant – albeit perhaps somewhat incremental - scientific advance.

 Minor points:

 Tables 6 and 7 seem too large for main text – it would be appropriate for them to be supplemental.

Authors response: we agree and converted both tables to supplemental files (New tables S6 and S7).

In Figure 4, it would be helpful to include the traditional names of human polyomaviruses – perhaps in parentheses after the accession number. Also note that it is unclear whether the virus initially dubbed “Lyon-IARC polyomavirus” should be re-named “Human polyomavirus 14” https://www.ncbi.nlm.nih.gov/pubmed/31320414 .

Authors response: All traditional names are listed in the old table S4 (now new table S8). We feel that additionally including the traditional names in Figs 4 and 5 would overload them and would exceed the available width. Therefore we propose to leave the figures as they are. Lyon-IARC polyomavirus is among a bundle of polyomaviruses which were proposed as species in 2019. This proposal is currently evaluated by the ICTV executive committee. The propsed species name is Human polyomavirus 14.

Lines 458-467 will be confusing to some readers. At face value it seems to first say that extensive cospeciation was detected but then tests for deep splits in host phylogeny did not detect any codivergence patterns. The authors should find a clearer way to explain the results. Discussion text starting at line 554 is somewhat clearer.

Authors response:  We tried to make our point clearer by modifying this paragraph. We think that it now better explains that (1) overall there is a signal for codivergence in the family and Alphapolyomavirus genus tree but (ii) most codivergence events happened during PyV diversification within host orders, with very little to no evidence pointing at deeper nodes in the PyV phylogeny mirroring the ordinal diversification of their hosts.

Line 518: Merkel cell polyomavirus lacks a MTAg. Instead, its ALTO gene – which theoretically resembles the second exon of MTAg – is thought to be translated through an internal ribosome entry or translational reinitiation event. The failure to detect devoted MTAg transcripts could simply be an indication that the virus resembles the organization of MCPyV/ALTO instead of murine polyomavirus 1/MTAg.

Authors response: We thank the reviewer for this helpful comment. We added in this paragraph the follwing: „In addition, the failure to detect MTAg transcripts could be an indication that these viruses resemble the organization of MCPyV/ALTO instead of murine polyomavirus 1/MTAg. For these novel viruses (except ClupPyV1) an ALTO ORF was  indeed predicted which theoretically resembles the second exon of MTAg. „ (lines 516-519 of the revised manuscript). „

Reviewer 2 Report

Manuscript by Ehlers et al. (viruses-594523) described discovery of 11 new mammalian polyomaviruses, identification of their coding genes and phylogenetic analyses, and proposition of renewed polyomaviral classification based on their findings. To extend our knowledge of polyomaviruses associate with diverse mammalian species they examined “1614 samples from 1222 animals of 44” mammals. Using methods established by the authors’ group they also revealed glimpse of complex splicing variations of polyomaviral early regions. Although data presentation of such laborious work especially needs self-explanatory organization, which in some extent lacked in this manuscript, their works undoubtedly shed light on unexplored era of polyomaviral diversity. With minor modifications, this reviewer feels that the manuscript is suitable for publication in Viruses.

Specific points,

In Table 4, this reviewer disagrees with the list of late genes of Philantomba monticola polyomavirus 1 as if it encodes agnoprotein (synonym of Vp2 ORF, as cited in Text S6). The 92 a.a. ORF referred to as Vp2 uORF is encoded in the reverse orientation to the late genes. All agnoproteins (or Vp2 uORFs) are encoded in the forward orientation with late genes.

Table 7 is highly confusing. Simply showing tiny marks representing “methods used for identifying splicing sites” and almost incomprehensible description of parenthesis markings are unacceptable. Presentation requires modification for clarifying what was experimentally proven or suggested by phylogenetic and/or in silico analysis. Depicting schematics as example could help readers to understand what were described.

Minor points.

In line 198, “Based on these sites, annotation of CDS was - if necessary - corrected.” Please elaborate how it was corrected.

In line 296, “To ascertain the natural host…”.

Although the detection of the virus in the multiple samples from the subject animal strongly suggested that the animal was the “natural host”, there is a fine line between association of the virus and natural host to the virus. The described method would not “ascertain the natural host” of the virus.

Author Response

VIRUSES_594523

Reviewer 2

Comments and Suggestions for Authors

Manuscript by Ehlers et al. (viruses-594523) described discovery of 11 new mammalian polyomaviruses, identification of their coding genes and phylogenetic analyses, and proposition of renewed polyomaviral classification based on their findings. To extend our knowledge of polyomaviruses associate with diverse mammalian species they examined “1614 samples from 1222 animals of 44” mammals. Using methods established by the authors’ group they also revealed glimpse of complex splicing variations of polyomaviral early regions. Although data presentation of such laborious work especially needs self-explanatory organization, which in some extent lacked in this manuscript, their works undoubtedly shed light on unexplored era of polyomaviral diversity. With minor modifications, this reviewer feels that the manuscript is suitable for publication in Viruses.

Specific points,

In Table 4, this reviewer disagrees with the list of late genes of Philantomba monticola polyomavirus 1 as if it encodes agnoprotein (synonym of Vp2 ORF, as cited in Text S6). The 92 a.a. ORF referred to as Vp2 uORF is encoded in the reverse orientation to the late genes. All agnoproteins (or Vp2 uORFs) are encoded in the forward orientation with late genes.

Authors response:

We have clarified in Text S6 (New Text S10) and now write as last paragraph:

„ORF upstream of the VP2 gene. We identified ORFs (> 49 aa) upstream of the VP2 gene in 6/11 PyV genomes (CaegPyV1, GgliPyV1, PleoPyV1, PmonPyV1, PporPyV1, TgliPyV1; Table 4) encoding a putative protein of unknown function. These uORFs are in the same orientation (PmonPyV1: reverse orientation) to the VP2 coding sequence. In BKPyV, JCPyV, and some closely related primate PyVs, an additional agnoprotein coding sequence is found upstream of the VP2 coding sequence, in the same orientation. Agnoprotein is a multifunctional polypeptide implicated in viral transcription, replication, assembly, maturation and release [11]. However, all putative proteins which are encoded by the uORFs identified here, lack similarity to agnoproteins (not shown).”

 We have also rephrased the foonote a in Tab. 4: „ a uORF: upstream ORF (function of encoded protein unknown; no similarity to agnoproteins)“

Table 7 is highly confusing. Simply showing tiny marks representing “methods used for identifying splicing sites” and almost incomprehensible description of parenthesis markings are unacceptable. Presentation requires modification for clarifying what was experimentally proven or suggested by phylogenetic and/or in silico analysis. Depicting schematics as example could help readers to understand what were described.

Authors response: Table 7 has been converted into Supplementary file S7 in response to reviewers 1and 3. S7 is now an Excel file with more and wider columns. This allowed us to rephrase column contents and to list splice sites identified in the current study and those identified and published earlier in different columns. As a result, most footnotes have been deleted and parentheses omitted. Overall readability is improved.

Minor points.

In line 198, “Based on these sites, annotation of CDS was - if necessary - corrected.” Please elaborate how it was corrected.

Authors response: We clarified the description of our approach and wrote: „Based on the splice sites (i) experimentally identified here for 11 novel and 7 published polyomaviruses and (ii) experimentally identified for 10 published polyomaviruses by others [7, 32-43], predicted LTAg, MTAg, and STAg splice sites of 79 published, annotated polyomaviruses were checked in early region nucleic acid alignments of phylogenetically closely related viruses and if necessary corrected or newly selected. Criteria for selection of splice sites were conservation in position and sequence with those of the most closely related phylogenetic clade member which was experimentally examined. Based on these sites, annotation of CDS was - if necessary - corrected.“ Lines 197-203 of the revisend manuscript.

In line 296, “To ascertain the natural host…”.

Although the detection of the virus in the multiple samples from the subject animal strongly suggested that the animal was the “natural host”, there is a fine line between association of the virus and natural host to the virus. The described method would not “ascertain the natural host” of the virus.

Authors response: We can indeed not prove with our data that the host is the natural i.e. the original host. The detection of a PyV in a certain host may have been preceeded by horizontal transmission from the „natural/original“ host to a new host. To clarify, we omitted the term „natural“ and wrote: „To further evaluate the potential host association of those of the 11 novel, completely sequenced polyomaviruses where only 1 or 2 of the tested samples were positive in generic PCR, virus genome-derived specific primers were selected and used in nested PCR for more sensitive screening. This was successful for 7 of the 8 tested polyomaviruses, thereby increasing the number of positive samples and individuals.“ Lines 290-294 of the revisend manuscript.

Reviewer 3 Report

            In the article "Novel polyomaviruses in mammals from multiple orders and reassessment of polyomavirus evolution and taxonomy", Ehlers et al. aimed at a deeper insight into the evolution of polyomaviruses by identifying them in host orders and families that have rarely or not been studied. They identified 16 novel polyomaviruses and fully sequenced 11 of them. They also predicted and examined by molecular biology the splice sites for large and small T antigen (LTAg, STAg) coding sequences (CDS) and they performed co-phylogenetic analyses based on LTAg CDS, which suggested that an update of the taxonomy of the family is required.

            Overall the experimental quality of the study is good. My major concern relates to the length of the manuscript (49 pages). I am aware that the authors performed a huge work but I think that the quality of the manuscript would be improved if it was shortened. At least, the tables 6 and 7, which are rather indigestible (7 pages each), could be added as supplementary materials.

Minor comments :

L 248 : “1222 animals” instead of “1228 animals” ?

For more clarity, “‐:‐” should be replaced by the taxon, the country and known or novel, in the tables 1 and 2.

L 296-301 : Several novel polyomaviruses where only 1 or 2 of the tested samples were positive in generic PCR, were not tested in nested PCR. For more clarity, it should be specified “This was successful for 7 of 8 polyomaviruses that have been tested”.

In the explanatory box of the figures 1, 2 and 3, the authors should specify what is colored in gray.

The use of “detected” in the parts related to the identification of splice sites is confusing.
For instance :
L 307 : “The predicted LTAg introns were experimentally detected for each of the 11 novel polyomaviruses”.
L 344 : “STAg and LTAg‐encoding encoding sequences that were detected but not experimentally identified”.
It should be better to only use “predicted” and “experimentally identified”.

L 344 : “encoding encoding”

Author Response

VIRUSES_594523

Reviewer 3

Comments and Suggestions for Authors

            In the article "Novel polyomaviruses in mammals from multiple orders and reassessment of polyomavirus evolution and taxonomy", Ehlers et al. aimed at a deeper insight into the evolution of polyomaviruses by identifying them in host orders and families that have rarely or not been studied. They identified 16 novel polyomaviruses and fully sequenced 11 of them. They also predicted and examined by molecular biology the splice sites for large and small T antigen (LTAg, STAg) coding sequences (CDS) and they performed co-phylogenetic analyses based on LTAg CDS, which suggested that an update of the taxonomy of the family is required.

            Overall the experimental quality of the study is good. My major concern relates to the length of the manuscript (49 pages). I am aware that the authors performed a huge work but I think that the quality of the manuscript would be improved if it was shortened. At least, the tables 6 and 7, which are rather indigestible (7 pages each), could be added as supplementary materials.

Authors response: we agree and converted both tabs to supplemental files (New tables S6 and S7). To shorten the main text further, we also converted Tables 4 and 5 to supplemental files (New tables S3 and S4). This reduced the length oft he manuscript from 49 to 32 pages.

Minor comments :

L 248 : “1222 animals” instead of “1228 animals” ?

Authors response: we corrected the animal number

For more clarity, “‐:‐” should be replaced by the taxon, the country and known or novel, in the tables 1 and 2.

Authors response: All “‐:‐” symbols were exchanged against the respective terminus in Tabs. 1 and 2.

L 296-301 : Several novel polyomaviruses where only 1 or 2 of the tested samples were positive in generic PCR, were not tested in nested PCR. For more clarity, it should be specified “This was successful for 7 of 8 polyomaviruses that have been tested”.

Authors response: As specific nested PCR could only be designed for those novel polyomaviruses, for which full genome could be amplified and sequenced, specific nested PCR could only be performed for 8 novel polyomaviruses, and this was done for each of the 8 virueses. To clarify we wrote: „To further evaluate the potential host association of those of the 11 novel, completely sequenced polyomaviruses where only 1 or 2 of the tested samples were positive in generic PCR, virus genome-derived specific primers were selected and used in nested PCR for more sensitive screening. This was successful for 7 of the 8 tested polyomaviruses, thereby increasing the number of positive samples and individuals.“

In the explanatory box of the figures 1, 2 and 3, the authors should specify what is colored in gray.

Authors response: We added this information to the box in each figure 1, 2, and 3

The use of “detected” in the parts related to the identification of splice sites is confusing.
For instance :
L 307 : “The predicted LTAg introns were experimentally detected for each of the 11 novel polyomaviruses”.
L 344 : “STAg and LTAg‐encoding encoding sequences that were detected but not experimentally identified”.
It should be better to only use “predicted” and “experimentally identified”.

Authors response: we changed all respective phrasings accordingly

L 344 : “encoding encoding”

Authors response: corrected